# Performance of Fine-Tuning Convolutional Neural Networks for HEp-2 Image Classification

**Vincenzo Taormina [1], Donato Cascio [2,\*] , Leonardo Abbene [2] and Giuseppe Raso [2]**

1   Department of Engineering, University of Palermo, 90128 Palermo, Italy; vincenzo.taormina@unipa.it
2   Department of Physics and Chemistry, University of Palermo, 90128 Palermo, Italy;
    leonardo.abbene@unipa.it (L.A.); giuseppe.raso@unipa.it (G.R.)
\*   Correspondence: donato.cascio@unipa.it

**Abstract:** The search for anti-nucleus antibodies (ANA) represents a fundamental step in the diagnosis of autoimmune diseases. The test considered the gold standard for ANA research is indirect immunofluorescence (IIF). The best substrate for ANA detection is provided by Human Epithelial type 2 (HEp-2) cells. The first phase of HEp-2 type image analysis involves the classification of fluorescence intensity in the positive/negative classes. However, the analysis of IIF images is difficult to perform and particularly dependent on the experience of the immunologist. For this reason, the interest of the scientific community in finding relevant technological solutions to the problem has been high. Deep learning, and in particular the Convolutional Neural Networks (CNNs), have demonstrated their effectiveness in the classification of biomedical images. In this work the efficacy of the CNN fine-tuning method applied to the problem of classification of fluorescence intensity in HEp-2 images was investigated. For this purpose, four of the best known pre-trained networks were analyzed (AlexNet, SqueezeNet, ResNet18, GoogLeNet). The classifying power of CNN was investigated with different training modalities; three levels of freezing weights and scratch. Performance analysis was conducted, in terms of area under the ROC (Receiver Operating Characteristic) curve (AUC) and accuracy, using a public database. The best result achieved an AUC equal to 98.6% and an accuracy of 93.9%, demonstrating an excellent ability to discriminate between the positive/negative fluorescence classes. For an effective performance comparison, the fine-tuning mode was compared to those in which CNNs are used as feature extractors, and the best configuration found was compared with other state-of-the-art works.

**Keywords:** CNNs; autoimmune diseases; IIF test; HEp-2; deep learning; fine-tuning; features extractors

## 1. Introduction

Autoimmune diseases occur whenever the immune system is activated in an abnormal way and attacks healthy cells, instead of defending them from pathogens; thus, causing functional or anatomical alterations of the affected district. [1]. There are at least 80 types of known autoimmune diseases (AD) [1]. AD can affect almost any part of the body i.e., tissue, organ, body system [2]. The anti-nucleus autoantibodies (ANA) are directed towards distinct components of the nucleus and are traditionally sought after with the indirect immunofluorescence (IIF) technique. With this method, in addition to the antibodies directed towards nuclear components, antibodies directed towards antigens are also highlighted and located in other cellular compartments. When Human Epithelial type 2 (HEp-2) cells are used as a substrate, the IIF method allows the detection of autoantibodies to at least 30 distinct nuclear and cytoplasmic antigens [3]. For this reason, the IIF technique with HEp-2 substrate is

considered the gold standard test for the diagnosis of autoimmune diseases. The patterns with which fluorescence can occur are associated with groups of autoimmune diseases.

In the IIF image analysis flow, the first phase concerns the fluorescence intensity classification. This phase is obviously of considerable importance in the diagnostic workflow, as it aims to discriminate on the presence/absence of autoimmune diseases. Figure 1 shows examples of positive/negative class.

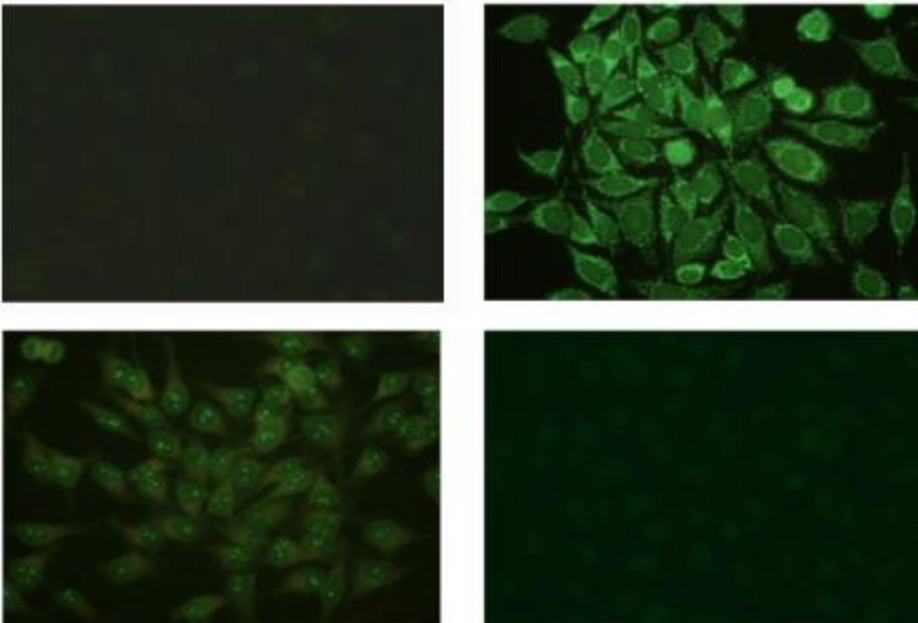

**Figure 1.** Examples of indirect immunofluorescence (IIF) images with different fluorescence intensity: in the upper left a positive centrioles image, in the upper right a positive cytoplasmic image, in the lower left a positive nucleolar image, in the lower right a negative image. The positivity/negativity of fluorescence is not strictly correlated with the luminous intensity of the Human Epithelial type 2 (HEp-2) images; the image in the upper left for example, although presenting low light intensity, has a positive fluorescence intensity.

However, the analysis of IIF images, as a whole, and in particular, in regards to the analysis of intensity, is extremely complex and linked to the experience of the immunologist [4]. For this reason, in recent years there have been numerous scientific works aimed at obtaining automatic support systems for HEp-2 image analysis [5,6]. In particular, in the last few years, various research groups interested in the topic have tried to exploit the potential offered by recent machine learning techniques in order to address the problem of the classification of HEp-2 images. However, the topic of fluorescence intensity classification is still poorly addressed [7].

Over the past decade, the popularity of methods that exploit deep learning techniques has considerably increased, evidently as deep learning has improved the state-of-the-art methods in research fields, such as speech recognition and computer vision [8]. Although neural networks had their scientific boom already in the 80s, their recent success can be attributed to an increased availability of data, improvements in hardware/software [9], and new algorithms capable of both speeding up learning in the training phase and improving the generalization of new data [10].

In the field of computer vision, deep learning has expressed its potential in image processing thanks to Convolutional Neural Networks (CNNs). In recent years there have been several fields of use of CNNs that, nowadays, allow the development of important applications. Some examples are: the antispoofing of the face and iris [11], recognition of highway traffic congestion [12], the image steganography and steganalysis [13], the galaxy morphology classification [14], drone detection, and classification [15,16].

Conceptually, the CNNs are inspired by the visual system as proposed in the works of Hubel and Wiesel on cat and monkey visual cortices [17]. A CNN accepts an image directly as input and applies a hierarchy of different convolution kernels to it. The first layers allow extraction of elementary visual features, such as oriented edges, end-points, and corners, and are gradually combined with subsequent layers in order to detect higher-order features [18].

In the field of image classification in general, CNNs can be used, proposing ad-hoc architecture and proceeding with its training, or using pre-trained architecture. In the latter case, the training on specific data to the problem can be carried out from scratch, or through fine-tuning of a part of the parameters/weights of the pre-trained network.

Fine-tuning is known as transfer learning, as the knowledge of another problem is exploited to solve the object of the study. Furthermore, a pre-trained CNN architecture can be modified in its architecture before carrying out training or fine-tuning.

In our previous work [19], we started to tackle the problem of fluorescence intensity classification by means of CNNs, but we did it by limiting their use to feature extractors. In this study, we analyzed various known CNNs and their respective layers from which to extract the features that allowed classification by means of Support Vector Machine (SVM) classifiers. The advantage of this type of work is the simplicity of implementation (no retraining of the pre-trained networks must be carried out), the disadvantage is usually relatively lower performances than those obtained from the pre-trained networks with the fine-tuning method. This is confirmed in the recent review by Rahman et al. [7], in fact the authors, regarding our previous work, comment as follows: "It is believed that the adoption of features from a fine-tuned CNN model would further improve the performance of Cascio et al. (2019c)'s method ".

The importance of the topic, the fact that it has been little addressed by the scientific community, and the possibility of analyzing the effectiveness of pre-trained networks, training them with the performing mode of fine-tuning, were the motivations that led us to face the problem of fluorescence intensity classification in HEp-2 images.

## 2. Literature Review

Numerous contributions have been made in recent years by the scientific community in the classification of HEp-2 images, especially due to contests organized on the subject, such as ICPR 2012 [20], ICIP 2013 [21], and ICPR 2014 [22]. However, these competitions have only dealt with the theme of pattern classification, considering 6/7 classes of significant interest.

In Li and Shen's work [23], a deep cross residual network (DCR-Net) was proposed. The structure is similar to DenseNet, but the authors, in order to minimize the vanishing gradient problem, added cross-residual shortcuts between some layers to the original network. The advantage of the change is to increase the depth of the network by improving feature representation. They made a date augmentation for rotations. The database used is I3A Task 1 and MIVIA. The method showed a Mean Class Accuracy (MCA) of 85,1% on a private test of I3A database.

Manivannan et al. [24] proposed a method that allowed them to win the ICPR 2014 contest with an MCA of 87.1%. They extracted a local feature vector that is aggregated via sparse coding. The method uses a pyramidal decomposition of the cell analyzed in the central part and in the crown that contains the cell membrane. Linear SVMs are the classifiers used on the learned dictionary. Specifically, they use four SVMs, the first one trained on the orientation of the original images, and the remaining three, respectively, on the images rotated by 90/180/270 degrees.

In a few cases, the authors have ventured into unsupervised classifications methods for the analysis of HEp-2 images [25,26]. In these cases, the number of analyzed patterns was reduced. In [25], the authors compared the method called BoW (Bag of Words), based on hand-crafted features and the deep learning model as learning hierarchical features, by clustering unsupervised learning. The authors obtained the best results with the BoW model. They demonstrated that, while the deep learning model allows better extraction of global features, the BoW model is better at extracting local

features. In [26], the authors proposed the grouping of centromeres present on the cells through a clustering K-means algorithm; the method allows the identification and classification of the centromere pattern. The method is tested on two public databases, MIVIA and Auto-Immunité, Diagnostic Assisté par ordinateur (AIDA), on which accuracy performances of 92% and 98%, respectively, are obtained. Despite the high indexing performance of the centromeric pattern, being able to classify only one type of pattern is certainly a limitation for the method.

Only recently has deep learning been applied to the various fields of HEp-2 image analysis [7]. In this context, one of the first works on the classification of HEp-2 images in which CNN was used is due to Gao et al. [27]. In this work, a CNN having eight layers, compared to the Bag of Features and Fisher Vector methods, was compared. The proposed CNN architecture has six convolution-type layers with pooling operations, and two fully-connected layers. The database used for the test is I3A Task 1 and MIVIA. The authors obtained 96.76% of mean class accuracy (MCA) and 97.24% of accuracy.

In [28], the authors used well-known CNNs in the pre-trained mode as vector feature extractors and trained multiple SVM classifiers in the one-against-all (OAA) mode. This strategy is one of the best known for breaking down a multi-class problem into a series of binary classifiers. Instead of using the classic OAA scheme, in which the test is assigned to the majority classifier result, the authors proposed the use of a K-nearest neighbors (KNN) based classifier as a collector of the various ensemble results. The performances on the public database I3A are in terms of MCA 82.16% and 93.75% at the cell level, and at the image level, respectively.

Some research groups have dealt with other aspects in the field of HEp-2 images, such as segmentation [29] and the search for mitosis [30].

The problem of classifying the intensity of fluorescence has certainly been little addressed [7]. The reason is probably due to the lack of public databases containing both positive and negative images; to date, it seems that the only public database of HEp-2 images with these characteristics is AIDA [4].

In Merone et al. [31], the problem of fluorescence intensity analysis was addressed but the authors did not make a classification between positive and negative, rather between positive, negative and weak positive. The authors extracted features through an Invariant Scattering Convolutional Network and used SVM classifiers with a Gaussian kernel. The network was based on multiple wavelet module operators and was used as a texture description. To classify the tree-class the authors applied the one-on-one approach and trained the tree binary classifier (negative vs. positive/negative vs. weak positive/negative vs. weak positive). Their method was trained on a private database of 570 wells for a total of 1771 images in which the fluorescence intensity was blindly classified by two doctors. The accuracy reported in the private database was 89.5%.

A classification of fluorescence intensity into positive vs. weak positive was also carried out by Di Cataldo et al. [32]. The authors used local contrast features at multiple scales and a KNN classifier to characterize the image, thereby achieving an accuracy of 85% in fluorescent intensity discrimination.

In Bennamar Elgaaied et al. [4] the authors have implemented a method based on SVM to classify the HEp-2 images in positive or negative intensity. They achieved an accuracy of 85.5% using traditional features based on intensity, geometry, and shape. The same set of tests was analyzed by two young immunologists verifying a greater ability of the automatic system (85.5% vs. 66%).

Other authors addressed the classification in positive/negative intensity on private databases. Iannello et al. [33] used a private database with 914 images to train a classifier able to discriminate between positive and negative intensity. Some areas of interest, called patches, have been extracted from the training set with the aid of the scale-invariant feature transform (SIFT) algorithm; therefore, 19 features are extracted from these. The features were extracted from first and second order gray level histograms. Two method of feature reduction was applied, the principal component analysis (PCA) and the linear discriminant analysis (LDA). Finally, the classification was based on the Gaussian mixture model and reaches an accuracy of 89.49%.

In the work of Zhou et al. [34] the authors presented a fluorescence intensity classification method in which private databases are analyzed. The method makes use of the fusion of global and local type

features. For simple cases, they use the SVM classifier with global features, while for doubtful cases, they proposed a further classification based on local features combined with another SVM. The results show an accuracy of 98.68%. However, as the analysis was conducted on a private database, the work does not allow easy performance comparison.

Despite its recognized potential, deep learning, and, in particular, the fine-tuning training method, has hardly been used for fluorescence intensity analysis. Only in one of our recent works presented at congress [35], was the fine-tuning method used with the pre-trained GoogLeNet network. In this study, albeit with a very limited analysis, promising results were obtained. In fact, the accuracy was 93% with an area under the Receiver Operating Characteristic (ROC) curve (AUC) of 98.4%. A synthetic representation of the literature presented here, with its pros and cons, is shown in the Table 1.

In the present work we analyzed the effectiveness of the fine-tuning training method applied to the problem of fluorescence intensity classification in HEp-2 images. For this purpose four of the best known pre-trained networks (AlexNet, SqueezeNet, ResNet18, GoogLeNet) have been analyzed. Different training modalities have been investigated; three levels of freezing weights and scratch. The effect that data augmentation by rotations can lead to classification performance was tested and the k-fold cross validation procedure was applied to maximize the use of the database examples. For an effective performance comparison, the fine-tuning mode was compared with those in which CNNs are used by feature extractors. The best configuration found was compared with other state-of-the-art works.

**Table 1.** Literature related to automatic analysis of HEp-2 images.

| Reference | Method and Database | Pros | Cons | Purpose of the Research |
|---|---|---|---|---|
| Percannella [29] | -Preprocessing with histogram equalization and morphological operations. Double segmentation phase based on the use of a classifier. -MIVIA dataset. | Accurate segmentation masks. A public database was used. | The analysis was conducted on only six cell patterns. Mitoses were not considered. | HEp-2 cells segmentation. |
| Gupta [28] | -Data augmentation. Linear Support Vector Machine (SVM) trained with features extracted from AlexNet. -I3A dataset. | It is one of the few works for the classification of mitosis. A public database was used. | A classification phase of fluoroscopic patterns, to verify whether the identification of mitosis allows a substantial improvement in performance, is missing. | HEp-2 cells classification into mitotic and non-mitotic classes. |
| Shen [23] | -Preprocessing with stretching and data augmentation. Classification based on Deep Co-Interactive Relation Network (DCR-Net). -I3A task1 and MIVIA dataset. | The implemented method proved to be robust and performing by winning the ICPR 2016 contest on the I3A task1 dataset. | The classification is cell level based, but the cells was manually segmented. | Classification of (six) staining patterns. |
| Manivannan [24] | -Preprocessing with intensity normalization. Pyramidal decomposition of the cells in the central part and in the crown. Four types of local descriptors as features and sparse coding as aggregation. Classification based on linear SVM classifiers. -I3A task1 and I3A task2 dataset. | The pyramidal decomposition proved to be very efficient, in fact the method won the ICPR 2014 contest on the I3A task1 and task2 dataset. | The final classification is obtained considering the maximum value of the classifiers used. The use of an additional classifier could give greater robustness and better performance. | Classification of (six/seven) staining patterns with/without segmentation masks. |
| Gao [25] | -Preprocessing with intensity normalization. Bag of Words based on scale-invariant feature transform (SIFT) features, comparing with deep learning model (Single-layer networks for patches classification and multi-layer network for full images classification). Classification by k-means clustering. -I3A task1 and MIVIA dataset. | A comparison between the traditional method based on Bag of Words and the method based on deep learning was made. | The classification is cell level based, but the cells was manually segmented. | Classification of (six) staining patterns. |
| Vivona [26] | -Preprocessing with contrast limited adaptive histogram equalization (CLAHE) and morphological operations like dilatation and holes filling. Automatic segmentation of the Centromere pattern. Classification by k-means clustering. -AIDA and MIVIA dataset. | Good performance of centromere identification without manual segmentation and supervised dataset. A public database was used. | Only one fluorescence pattern was analyzed. | Classification of centromere pattern. |
| Gao [27] | -Preprocessing with intensity normalization and data augmentation. CNN (Convolutional Neural Network) with eight layers (six convolutional layers and two fully connected layers). -I3A task1 and MIVIA dataset. | A comparison between the traditional methods such Bag of words and Fisher Vector (FV) was made. | The classification is cell level based, but the cells was manually segmented. | Classification of (six) staining patterns. |
| Cascio [28] | -Preprocessing with stretching and data augmentation. Six linear SVM trained with features extracted from CNN. Final classification with KNN (K-nearest neighbors) to improve classic one-against-all strategy. -I3A task1 dataset. | An intensive analysis of procedures and parameters was conducted, which allowed performances among the highest on the I3A task1 public database. | The classification is cell level based, but the cells was manually segmented. | Classification of (six) staining patterns. |

**Table 1.** *Cont.*

| Reference | Method and Database | Pros | Cons | Purpose of the Research |
|---|---|---|---|---|
| Di Cataldo [32] | -Preprocessing based on histogram equalization and morphological opening. Segmentation with the Watershed technique. KNN classifier with morphological features and global/local texture descriptors to classify the patterns and KNN with local contrast features at multiple scale to classify intensity fluorescence. <br> -I3A and MIVIA dataset. | One of the most comprehensive works for HEp-2 image classification. | Probably due to the database at their disposal, they do not analyze the positive/negative fluorescence intensity. | Classification between positive and weak positive fluorescence classes. Classification of (six) staining patterns. |
| Merone [31] | -Three SVM with Gaussian kernel trained on features extracted from Invariant Scattering Convolutional Network based on wavelet modules. Final classification based on one-against-one strategy. <br> -Private dataset. | The network was based on multiple wavelet module operators and was used as a texture description, in this way it was particularly effective. | The use of a private dataset of HEp-2 images for training/test the method. | Classification between positive, negative and weak positive fluorescence classes. |
| Bennamar [4] | -Separate preprocessing for each class to recognize. Traditional features extraction and separate features reduction for each class. Classification based on seven SVM with Gaussian kernel. Final classification with KNN. <br> -AIDA database. | The only work published in which a complete Computer aided diagnosis (CAD) system is presented to classify HEp-2 images in terms of fluorescence intensity and florescence pattern. The CAD performance is comparable with the Junior immunologist. | Perfectible classification performance. | Fluorescence intensity classification and classification of (seven) staining patterns. |
| Iannello [33] | -SIFT algorithm to detect patches and features extracted from first and second order gray level histograms. Features selection with Principal Component Analysis (PCA) and Linear Discriminant Analysis (LDA). Final classification with Gaussian mixture model. <br> -Private dataset. | The features reduction process was particularly careful | A private database was used. | Fluorescence intensity classification |
| Zhou [34] | -An adaptive local thresholding was applied to cell image segmentation. <br> Two step classification: (1) with global features based on mean and variance of each channel RGB; (2) with local features based on SIFT and bag of words for doubt cases. <br> -Private dataset. | The classification process is very effective because it is trained on the segmented cells (without the background). | A private database was used. | Fluorescence intensity classification |
| Cascio [19] | -Data augmentation. Pre-trained CNNs dataset used as features extractors coupled with linear SVM. <br> -AIDA dataset. | An intense analysis of the parameters involved was conducted in order to maximize performance. | Convolutional neural networks were used only as feature extractors. | Fluorescence intensity classification. |
| Taormina [35] | -Data augmentation. The GoogLeNet network was used, both as a feature extractor and training it with fine-tuning mode. <br> -AIDA dataset. | The use of fine tuning has proved very promising. | Limited configurations explored. | Fluorescence intensity classification. |

## 3. Materials and Methods

### 3.1. Database and Cross-Validation Strategy

The analysis conducted in this work made use of the data contained in the public AIDA database [4].

Starting in November 2012, the Auto-Immunité, Diagnostic Assisté par ordinateur (AIDA) project is an international strategic project funded by the European Union (EU) in the context of ENPI Italy–Tunisia cross-border cooperation, in which one of the objectives was to collect a large database available to the scientific community.

Using a standard approach, seven immunology services have collected images of the IIF test on HEp-2 cells accompanied by the report of the senior immunologists. The public AIDA database consists of images only where three physician experts (independently) have expressed a unanimous opinion when reporting.

The AIDA database consists of 2080 images relating to 998 patients (261 males and 737 females); of these images, 582 are negative, while 1498 show a positive fluorescence intensity. The AIDA database is the public HEp-2 image database containing both images with positive fluorescence intensity and negative images; the other public databases are essentially composed of positive and weak positive fluorescence images, but not negative cases.

Besides being "numerous", the database is extremely varied, containing fluorescence positive sera with a variety of more-than-twenty staining patterns. In each image, a single or multiple pattern can be present. The pattern terminology is in accordance with the *International Consensus on ANA Patterns (ICAP)*. Available online: http://www.anapatterns.org (accessed on 8 September 2020) [36]. Moreover, manufacturers of kits and instruments employed for the ANA testing were different site-to-site, as well as, different automated systems solutions for the processing of Indirect Immunofluorescence tests have been used: IF Sprinter Euroimmun, NOVA from INOVA diagnostic, and Helios from Aesku. HEp-2 images have been acquired, after incubation of the 1/80 serum dilution, by means of a unit consisting of a fluorescence microscope (40-fold magnification) coupled with a 50 W mercury vapor lamp and a digital camera. The camera has a CCD sensor equipped with pixel size that equals 3.2 µm. The images have 24 bits color-depth and were stored in different common image file formats, as "jpg", "png", "bmp", and "tif". The public database can be downloaded, after registration, from the download section of the site AIDA Project. Available online: http://www.aidaproject.net/downloads (accessed on 8 September 2020).

In order to make maximum use of the available data, the k-fold validation technique at the level of the specimen was used for the training–validation–test chain in this work [24]. In fact, in order not to affect the performance, the images belonging to the same well were used entirely, for training, or for testing. According to this method, the database was divided into five folds (k = 5), with this strategy, five trainings and related tests were carried out. Regarding the composition of the sets, approximately each training was obtained using 20% of the dataset for the test, while the remaining 80% was divided into training and validation to the extent of approximately 64% and 16% of the dataset.

### 3.2. Statistics

When it is necessary to discriminate between two classes (which in the case of medical imaging problems often represent healthy/pathological classes), the evaluation of the performance of a diagnostic system is usually expressed through a pair of indices: sensitivity and specificity. In our case, the sensitivity represents the fraction of correctly recognized positive images, true positives (*TP*), on the total number of positive images, obtained by adding the true positives and the false negatives (*FN*), i.e.:

$$Sensitivity = \frac{TP}{TP + FN} \tag{1}$$

The specificity of a test is the fraction of recognized negative images, true negatives (*TN*), on the total of negative images, obtained by adding the true negatives and the false positives (*FP*), i.e.:

$$Specificity = \frac{TN}{TN + FP} \tag{2}$$

Using the aforementioned indices, various figures of merit recognized in medical imaging are obtained. One is certainly accuracy, which is defined as follows:

$$Accuracy = \frac{TP + TN}{TP + FN + TN + FP} \tag{3}$$

An automatic classification system generally returns an output value in the definition range; in this case between zero (image negativity) and one (image positivity). The output value must be compared with a predefined threshold value in order to associate the generic image with the positive/negativity classes. This leads to a variability of performance, in terms of the figures of merit introduced above, based on the threshold value chosen. For this reason, very often in medical statistics the Receiver Operating Characteristic (ROC) curve is used, which allows a performance analysis not dependent on the choice of the threshold value. In fact, when the threshold value changes, different pairs of sensitivity and specificity that will define the specific ROC. Therefore, a further figure of merit for classification systems is the area under the ROC curve, generally indicated with AUC.

### 3.3. Convolutional Neural Networks

The use of Deep Neural Networks for solving classification problems has grown significantly in recent years. In particular, Convolutional Neural Networks (CNN) are widely used [37]; the name derives from the convolution operations present in them. The success of deep learning and CNNs is certainly due, in addition to the high classification performance demonstrated by these classification methods, also to the ease of carrying out a classification process using these tools. In fact, the traditional chain composed of preprocessing, features extraction, training model, is entirely replaced by CNNs, which in their training process include feature extraction. CNNs are networks specialized in data processing which have the form of multiple vectors with a known grid-form topology. An example of this type of data can be a time series, which can be seen as a grid at a size sampled from regular intervals, or an image, which can be seen as a two-dimensional grid of pixels containing the intensity value for the three color channels (RGB).

The architecture of a CNN is structured as a series of representations made up of two types of layers: the convolution layers and the pooling layers (see Figure 2). The units in the convolution layers are organized into feature maps, in which each unit is connected to a local portion of the map of the next layer through a set of weights called a filter bank. The result of this local weighted average is then passed through a nonlinear function such as Rectified Linear activation Function (ReLU). All units in a feature map share the same filter bank.

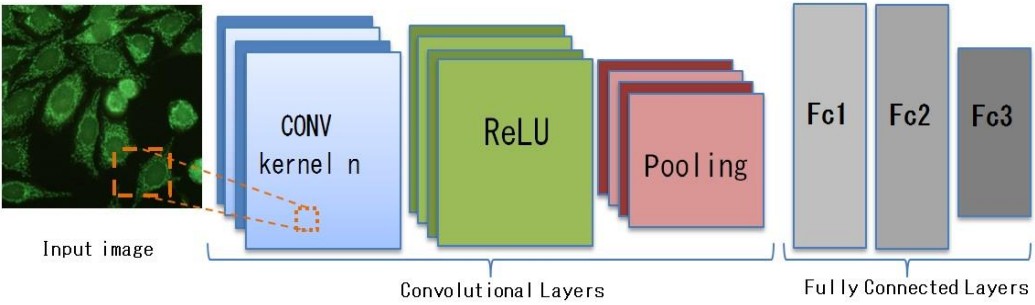

**Figure 2.** General scheme of the architecture of a the Convolutional Neural Network (CNN).

The capacity of a CNN can vary based on the number of layers it has. In addition to having different types of layers, a CNN can have multiple layers of the same type. In fact, there is rarely a single convolutional level, unless the network in question is extremely simple. Usually a CNN has a series of convolutional levels, the first of these, starting from the input level and going towards the output level, is used to obtain low-level characteristics, such as horizontal or vertical lines, angles, various contours, etc. The levels closest to the output level produce high-level characteristics, i.e., they represent rather complex figures such as faces, specific objects, a scene, etc. Since the design and training of a CNN is a complex problem, where exhaustive research cannot be used, and training requires large and accurate databases for training, and a computing power that is not always accessible to researchers, a well-established way in the literature is to use pre-trained CNN networks. Thanks to ImageNet competition, extremely powerful CNN networks have emerged [38]. These CNNs have been trained on over a million images to model generic feature rich representations.

In this work, CNNs were used to classify the fluorescence intensity in HEp-2 images. In detail, the performances of four of the most well-known pre-trained CNNs were analyzed applying them according to the fine-tuning learning approach. The strategy according to which pre-trained networks are used as feature extractors was also evaluated. In this case, the classification phase is usually carried out in combination with a classifier. For this purpose, we have used linear SVM classifiers in this work. The choice of best configuration was performed automatically, using the AUC as a figure of merit in a validation process based on the k-fold method.

### 3.4. Pre-Trained CNNs

Considering the state-of-the-art methods for CNNs, we have chosen the following four architectures:

- AlexNet [10]: the AlexNet network is one of the first convolutional neural networks that has achieved great classification successes. The winner of the 2012 Image-Net Large-Scale Visual Recognition Challenge (ILSVRC) competition, this network was the first to obtain more than good results on a very complex dataset such as ImageNet. This network consists of 25 layers, the part relating to convolutional layers sees 5 levels of convolution with the use of ReLU and (only for the first two levels of convolution and for the fifth) the max pooling technique. The second part of the CNN is composed of fully connected layers with the use of ReLU and Dropout techniques and finally by softmax for a 1000-d output.

- SqueezeNet [39]: in 2016 the architecture of this CNN was designed to have performances comparable to AlexNet, but with fewer parameters, so as to have an advantage in distributed training, in exporting new models from the cloud, and in deployment on a FPGA with limited memory. Specifically, this network consists of 68 layers with the aim of producing large activation maps. The filters used instead of being $3 \times 3$ are $1 \times 1$ precisely to reduce the computation by 1/9. CNN is made up of blocks called "fire modules", which contain a squeeze convolution layer with 1x1 filters and a expand layer with a mix of $1 \times 1$ and $3 \times 3$ convolution filters. This CNN has an initial and a final convolution layer, while the central part of the CNN is composed of 8 fire module blocks. No fully connected layers are used but an average pooling before the final softmax.

- ResNet18 [40]: this CNN, introduced in 2015, and inspired by the connection between neurons in the cerebral cortex, uses a residual connection or skip connections, which jump over some layers. With this method, it is possible to counteract the problem of degradation of performance as the depth of the net increases, in particular the "vanishing gradient". This CNN is made up of 72 layers, the various convolutions are followed by batch normalization and ReLU, while the residual connection exploits an additional layer of two inputs. The last layers consist of an average pooling, a fully connected layer and softmax.

- GoogLeNet [41]: this architecture is based on the use of "inception modules", each of which includes different convolutional sub-networks subsequently chained at the end of the module. This network is made up of 144 layers, the inception blocks are made up of four branches, the first three with $1 \times 1$, $3 \times 3$, and $5 \times 5$ convolutions, and the fourth with $3 \times 3$ max pooling. After that,

all feature maps at different paths are concatenated together as the input of the next module. The last layers are composed of an average pooling and a fully connected layers and the softmax for the final output.

### 3.5. Fine-Tuning Description

Fine-tuning is a transfer learning technique that focuses on storing knowledge gained while solving one problem and applying it to a different but related problem [42]. Since CNNs are composed of numerous layers and a huge number of parameters, e.g., AlexNet has 650 K neurons and 60 M parameters (as an example, a graphical representation of the AlexNet architecture is shown in Figure 3), the network training phase should benefit from the use of databases rich in examples that avoid the problem of overfitting. Unfortunately, it is not always possible to take advantage of such large databases. Furthermore, training a CNN effectively better on a large database is demanding both in terms of computation time and the computational resources required.

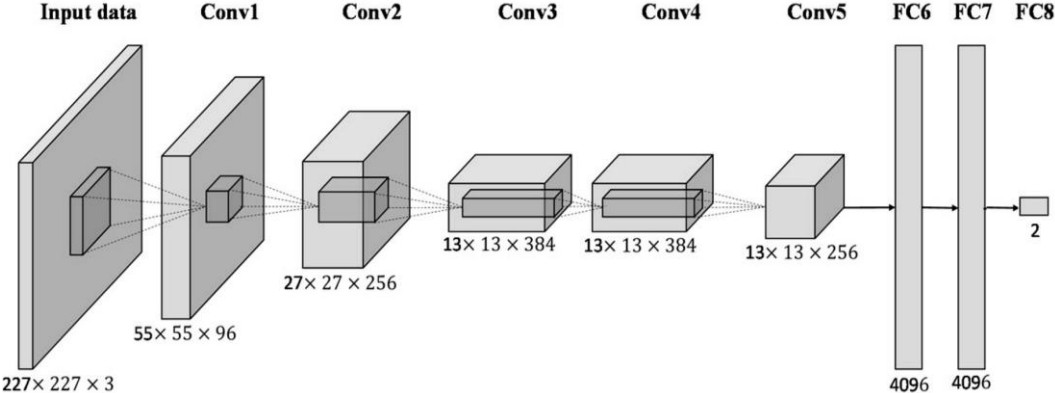

**Figure 3.** Schematic of the AlexNet architecture. The output has been replaced considering the two classes on which to discriminate.

Fine-tuning arises from the need to make up for these deficiencies. This method consists in the possibility of using a Neural Network, pre-trained on a large database, through a further training phase with another database, even a small one. The output level is replaced with a new softmax output level, adjusting the number of classes to the classification problem being faced. The initial values of the weights used are those of the pre-trained network, except for the connections between the penultimate and last level whose weights are randomly initialized. New training iterations (SGD) are performed to optimize the weights with respect to the peculiarities of the new dataset (it does not need to be large). Fine-tuning can be done in two ways. One way is to freeze the weights of some layers and carry out new training cycles to modify the weights of the remaining layers. The concept of fixing the weights of the layers is defined as freezing of the layers. Generally, they are the first layers to be frozen as the first layers capture low level features. The greater the number of frozen layers, the less the fine-tuning effort in terms of time and resources is required.

In this case, the weights of the first CNN levels are frozen and the remaining parameters/weights are trained. The other way is to have the architecture re-train entirely on the new database. This method is called training from scratch. It is intuitive that the greater the number of frozen layers the lower the computational cost of training, so training from scratch is the most expensive form of computational training.

### 3.6. Hyperparameter Optimization

The design of a generic CNN provides, in addition to establishing the architecture and the topology, the optimization of parameters and connection weights through optimization procedures in the training phase. It refers to the numerous values to be optimized as hyperparameters optimization.

The pre-trained CNNs have a consolidated and optimized architecture for the database for which they were originally trained.

The process of optimizing the parameters during the training phase is certainly not trivial. Among the most recognized methods, there is the Stochastic Gradient Descent with Momentum (SGDM) that minimizes the loss function at each iteration considering the gradient of the loss function on the entire training dataset. The momentum term reduces the oscillations of the SGDM algorithm along the path of steepest descent towards the optimum. The momentum is responsible for reducing some noise and oscillations in the high curvature regions of the loss function generated by the SGD. A variant of the SGD uses training subsets called mini-batches, in this case a different mini-batch is used at each iteration. Simply put, the mini-batch specifies how many images to use in each iteration. The full pass of the training algorithm over the entire training set using mini-batches is one epoch. SGDM with mini-batch was used in this work.

Another fundamental parameter of the training process is the learning rate that allows to set the learning speed of the training process with the level of improvement of the network's weights. Conceptually, a high learning rate increases the speed of training execution by sacrificing the performance of the trained network, while a low learning rate will increase the training times by optimizing the network's weights that will be more performing. This parameter defines the level of adjustments of weight connections and network topology applied at each training cycle. A small learning rate permits a surgical fine-tune of the model to the training data, at the cost of a greater number of training cycles and longer processing times. A high learning rate permits the model to learn more quickly, but may sacrifice its accuracy caused by the lack of precision over the adjustments. This parameter is generally set to 0.01, but in some cases, it is interesting to be fine-tuned, especially when it is necessary to improve the runtime when using SGD. Table 2 shows the search space analyzed for the optimization of the parameters.

**Table 2.** Hyperparameters grid search.

| Parameter | Configurations |
|---|---|
| Training mode | Stochastic Gradient Descent with Momentum with mini-batch |
| Mini-batch size | {4, 8, 16, 32, 64, 128, 256} |
| Learning Rate | {0.01, 0.001, 0.0001} |
| Momentum coefficient | 0.9 |
| Epoch | max 10 epoch if freeze some layers max 30 epoch if training CNN from scratch |

The extraction of features from a pre-trained CNN is simple to implement and computationally fast as it exploits the implicit power of representing CNN characteristics. In this work, the feature vector extracted from the generic CNN is used to train a SVM classifier. As the size of the feature vector is large, we have chosen to train a linear SVM that has one hyperparameter: the penalty parameter "C" of the error term. The search for linear kernel parameter "C" is carried out using MATLAB "logspace" function in the range $[10^{-6}, 10^{2.5}]$; twenty equidistant values on a logarithmic scale were analyzed.

### 3.7. Training Strategy and Classification

In this work the binary classification of HEp-2 images in positive and negative fluorescence intensity was analyzed. In this regard, the transfer learning technique was used on CNN pre-trained networks.

In detail, the fine-tuning approach was analyzed for each CNN network, according to which, starting from the generic pre-trained CNN, the parameters are optimized by carrying out a training using the new database. In general, to implement fine-tuning, the last layer must be replaced to

correctly define the number of classes to be discriminated. The problem analyzed in this work, since we want to discriminate between two classes, turns out to be binary.

In the present work, the four CNN networks described in Section 3.4 have been analyzed, training them both in scratch mode and with fine-tuning considering three different depths of freeze. Table 3 shows the three freeze levels chosen for each CNN.

**Table 3.** Number of frozen layers at different levels and for the CNN analyzed.

| CNN Name | Total Layers | Low Frozen | Medium Frozen | High Frozen |
|---|---|---|---|---|
| AlexNet | 25 | 9 | 16 | 19 |
| SqueezeNet | 68 | 11 | 34 | 62 |
| ResNet18 | 72 | 12 | 52 | 67 |
| GoogLeNet | 144 | 11 | 110 | 139 |

As an example, Figure 4 shows the 25 layers that make up the AlexNet CNN with the graphic overlay of the three frozen levels chosen to perform the fine-tuning. The first level called low frozen indicates that the weights of the first nine layers of CNN AlexNet are fixed with the values of the network pre-trained on the original ImageNet database. The fine-tuning in this case consists in applying training cycles to CNN by changing the weights of the remaining layers at each iteration, i.e., from layer 10 to the last. In a very similar way to the first level, the medium frozen and high frozen levels were taken into consideration, which, respectively, fix the weights of the pre-trained AlexNet network up to layer 16 and layer 19; the fine-tuning in these two cases is carried out by modifying the weights of the last 9 layers and 6 layers. As described in sub-Section 3.6, the three selected fine-tuning levels are analyzed by iterating on the learning rate values {0.01, 0.001, 0.0001} and on the various batch sizes considering a maximum of 10 epochs. The results obtained and the related best configurations are reported in the next section.

**Figure 4.** Example diagram of the three levels of freezing of the weights referred to the AlexNet layers.

Table 4 shows, by way of example, part of the code to carry out the fine-tuning of CNN AlexNet. The code is written using MATLAB R2020a [43] and conceptually consists of seven steps.

**Table 4.** Steps for the fine-tuning of the pre-trained AlexNet network.

| | |
|---|---|
| Step 1: load AlexNet pre-trained CNN and take input image size. After download and install Deep Learning Toolbox Model for AlexNet Network support package, set "myNet" as AlexNet CNN pre-trained on the ImageNet data set. set "inputSize" because AlexNet requires input images of size 227×227×3. | %%% load pre-trained network myNet = alexnet; %%% take input image size inputSize = myNet.Layers(1). InputSize; |
| Step 2: load images. Load the training/validation/test images with the imageDatastore function that automatically labels the images based on folder names and stores the data as an ImageDatastore object. The imageDatastore function is optimized for large image data and efficiently read batches of images during training. | %%% load training / validation / test imdsTrain = imageDatastore(dir_training), 'IncludeSubfolders',true, 'LabelSource', 'foldernames'); imdsValid = imageDatastore(dir_validation), 'IncludeSubfolders',true, 'LabelSource', 'foldernames'); imdsTest = imageDatastore(dir_test), 'IncludeSubfolders',true, 'LabelSource', 'foldernames'); |

**Table 4.** *Cont.*

| | |
|---|---|
| Step 3: resize images and apply augmentation with rotation. With the augmentedImageDatastore function, the training/validation/test images are resized and augmented with rotation of 20°. | %%% resize images and augmentation with rotation<br>imageAugmenter = imageDataAugmenter('RandRotation',(−20,20));<br>augimdsTrain = augmentedImageDatastore(inputSize(1:2), imdsTrain, 'DataAugmentation', imageAugmenter);<br>augimdsValid = augmentedImageDatastore(inputSize(1:2), imdsValid, 'DataAugmentation', imageAugmenter);<br>augimdsTest = augmentedImageDatastore(inputSize(1:2), imdsTest, 'DataAugmentation', imageAugmenter); |
| Step 4: replace final layers.<br>Replace final layers considering that the new classes are only positive and negative while the pre-trained AlexNet are configured for 1000 classes. | %%% replace final layers<br>layersTransfer = myNet.Layers(1:end-3);<br>layers = (layersTransfer fullyConnectedLayer (2) % two class: positive and negative softmaxLayer classificationLayer); |
| Step 5: freeze initial layers.<br>With the freezeWeights function, the first network layers chosen are frozen so the new training does not change these weights. | %%% freeze initial layers<br>freeze = 16; % the first 16 layers layers(1:freeze) = freezeWeights(layers(1:freeze)); |
| Step 6: training network.<br>"Option" set the various training parameters such as the optimization algorithm, the learning rate, etc. The function trainNetwork executes the training on the network "myNet" considering the options, the freeze weights and the validation images set. | %%% training options<br>options = trainingOptions('sgdm', . . . % sgd with momentum<br>'MiniBatchSize', 32, 'MaxEpochs', 10, . . . 'LearnRate', 0.001, . . . 'Momentum', 0,9, . . . 'ValidationData', augimdsValid');<br>%%% train network using the training and validation data<br>myNet = trainNetwork(augimdsTrain, layers, options); |
| Step 7: test network.<br>With the classify function, the network trained on the validation set is tested on the test set, and the performance measures are extracted, such as ACC and AUC. Perfcurve function find the AUC and then the ROC curve is drawn with plot function. | %%% test network fine tuned<br>[pred, probs] = classify(myNet, augimdsTest);<br>%%% extract performance<br>(X,Y,T,AUC) = perfcurve(imdsTest.Labels, probs, 'POS');<br>fprintf('ACC: %d\n', accuracy); %print ACC<br>fprintf('AUC: %d\n', AUC); %print AUC<br>figure, plot(X,Y,'r'); %ROC curve<br>accuracy = mean(pred == imdsTest.Labels); |

## 4. Results

This section reports the procedures carried out and the results obtained regarding the classification of the fluorescence intensity in the positive/negative classes of the HEp-2 images with the fine-tuning method. For a better evaluation of the results, a comparison was made with the results obtained by the same CNNs used by feature extractors and finally a comparison with other state-of-the-art methods.

For the four pre-trained CNN networks described in Section 3.4, the transfer learning technique applied to HEp-2 images was analyzed. The division of the available patterns was carried out with a 5-fold cross validation and for each of the 5 iterations, training, validation, and testing were performed. The training phase was optimized considering the AUC as a measure of merit. IIF images have been resized to 227 × 227 (for AlexNet and SqueezeNet networks) and 224 × 224 (for GoogLeNet and ResNet18 networks) to be provided as input to CNN; no preprocessing has been applied to the image. CNN networks want a 3-channel image input, for this reason, we have evaluated the results using both RGB IIF images and only the green channel and replicating it on R and B channels. The results favored the second choice, to which all of the analyses reported in this work were conducted.

Moreover, we evaluated the application of the data augmentation using the rotation with angles of 20°. Data augmentation is a very effective practice especially when the data set for training is limited, or as in our case, when some classes are not particularly represented in the set of examples. In this way an 18 times larger sample data set was obtained.

The effect of this data augmentation was valued quantitatively in terms of performance. Table 5 shows the best results obtained, in terms of AUC, from the four CNNs analyzed:

**Table 5.** Best AUC results.

| CNN Name | High Frozen | Medium Frozen | Low Frozen | Scratch | Low Frozen + Data Augmentation | Scratch + Data Augmentation |
|----------|-------------|---------------|------------|---------|-------------------------------|----------------------------|
| AlexNet | 97.20% | 97.93% | 97.82% | 98.00% | 98.08% | 98.02% |
| SqueezeNet | 97.96% | 98.39% | 98.55% | 98.38% | 98.63% | 98.46% |
| ResNet18 | 97.27% | 97.88% | 98.11% | 98.24% | 98.33% | 98.32% |
| GoogLeNet | 96.30% | 96.78% | 98.00% | 97.96% | 98.37% | 98.20% |

The results are shown, considering the three levels of freeze of the pre-trained networks' weights and considering the retraining from scratch. The last two columns of Table 5 show the results obtained by performing a data augmentation, respectively, with the low frozen and with the scratch. As expected, the training with less freezing allows a better adaptation of the CNN to the classification problem. As for the data augmentation procedure, the results show that the application of this procedure leads, in all cases, to a slight improvement in performance. For each of the four CNN networks, Table 6 shows the parametric configuration that obtained the best result.

**Table 6.** Best Hyperparameter.

| CNN | AUC | Learning Rate | Mini-batch | Epoch |
|-----|-----|---------------|------------|-------|
| AlexNet | 98.08% | 0.001 | 16 | 4 |
| SqueezeNet | 98.63% | 0.001 | 16 | 6 |
| ResNet18 | 98.33% | 0.001 | 16 | 8 |
| GoogLeNet | 98.37% | 0.01 | 32 | 7 |

Figure 5 shows the ROC curve relating to the best configuration obtained, that is SqueezeNet with low frozen level and with data augmentation.

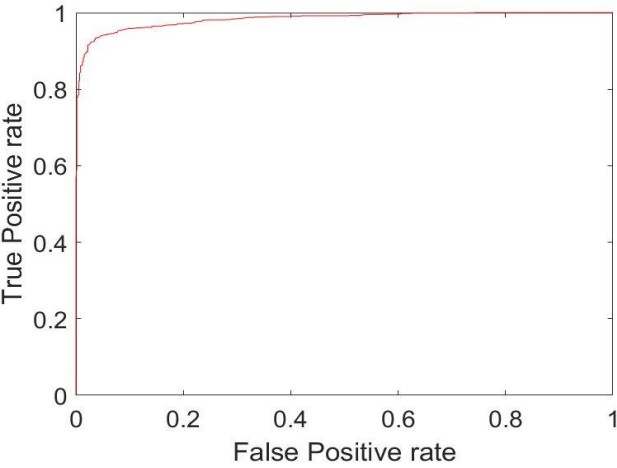

**Figure 5.** ROC (Receiver Operating Characteristic) curve of the best configuration.

In order to assess the significance of the results obtained with the CNNs trained with the fine-tuning technique, the performances obtained by the same CNN used as feature extractors were evaluated. In this case, the classification was achieved by means of linear SVM classifiers.

The idea on which the SVM is based is to find the best hyperplane that allows the "larger margin" between two classes represented by the projection in the n-dimensional space of the feature vector [44]. In the case of non-separable data, a hyperplane is sought that separates many, but not necessarily all the data, through the formulation of a "soft margin" using the penalty parameter C (or error term).

The linear SVM training phase was carried out by extracting the features, for the various layers, from all the training images. The SVM parameter C has been optimized on the validation set. The validated model was subsequently used for the test. Table 7 shows the best results obtained, in terms of AUC, for each pre-trained network analyzed and reports the layers that provided the best results.

**Table 7.** Best AUC results obtained by CNN used as feature extractors.

| CNN | AUC | Best Layers |
|---------|---------|---------------------|
| AlexNet | 95.52% | 'Conv 5' |
| SqueezeNet | 95.50% | 'Pool 10' |
| ResNet18 | 97.80% | 'Fc 1000' |
| GoogLeNet | 95.76% | 'Inception 3a output' |

The best result obtained with the CNNs used as a features extractor coupled to the SVM classifier reached an AUC equal to 97.8%. This result was obtained with the ResNet18 CNN. The SVM parameter C associated with the best configuration was C = 0.0298. The results in Table 7, when compared with those of Table 5, show that, with the same CNN used, the fine-tuning method achieves better performances than the other method.

Moreover, the results obtained in this work were compared with other state-of-the-art works in Table 8.

**Table 8.** Performance comparison.

| Method | Images Dataset | Accuracy | AUC |
|-----------------------|----------------|----------|--------|
| Iannello [32] | 914 | 89.49% | - |
| Bennamar Elgaaied [4] | 1006 | 85.5% | - |
| Zhou [34] | 1290 | 98.68% | - |
| Cascio [19] | 2080 | 92.8% | 97.4% |
| Taormina [35] | 2080 | 93.0% | 98.4% |
| Present method | 2080 | 93.93% | 98.63% |

Unfortunately, not all of the authors of these works in Table 8 calculated the AUC value. While, from the comparison in accuracy it is verified that our method turns out to be the second best. However, it should be noted that while Zhou et al. [34] aimed to maximize accuracy, the AUC was maximized in this work. For a more direct comparison with other methods that based their optimization on accuracy, we repeated the analysis using the latter index as a figure of merit, obtaining for the best configuration (also, this time with the SqueezeNet network), the result of 94.32% accuracy and an AUC equal to 98.34%.

It should also be noted that the AIDA database used in this work (as reported in sub-Section 3.1) is varied, both in terms of patterns contained (more than 20, both in single and multiple forms) and in terms of instruments and methods of acquisition (manufacturers of kits and instruments employed were different site-to-site); this makes it particularly difficult to classify.

*Running Time*

The training that provided the best performances required a calculation time of approximately 28 h for all 5 k-folds (using a 3.4 GHz Intel i7 CPU). Table 9 shows the calculation times, in terms of intervals obtained by varying the configuration hyperparameters (mini-batch, learning rate, epoch, etc.) necessary for the training of the various CNNs.

**Table 9.** CNNs training times.

| CNN Name | Training Time in Hours (min–max) | | | |
|---|---|---|---|---|
| | **High Frozen** | **Medium Frozen** | **Low Frozen** | **Scratch** |
| AlexNet | (3.42–9.57) | (4.33–12.58) | (4.69–17.24) | (5.65–26.3) |
| SqueezeNet | (6.26–10.22) | (7.43–14.15) | (11.93–27.96) | (12.56–36.57) |
| ResNet18 | (4.28–9.94) | (4.51–10.51) | (4.91–11.52) | (5.86–18.12) |
| GoogLeNet | (4.02–8.26) | (4.13–16.32) | (4.27–18.54) | (5.37–24.4) |

The configurations in which the CNNs were analyzed as feature extractors coupled with the SVM classifier required training times within the interval (0.57–13.6) hours.

## 5. Discussions and Conclusions

In this work, the discriminating capabilities of pre-trained CNNs applied to the problem of the classification of fluorescence intensity in HEp-2 images were investigated, a problem of crucial importance for the diagnosis of autoimmune diseases. To this end, four of the best known CNNs have been analyzed with the fine-tuning training method. In particular, the effectiveness of the fine-tuning method was verified in different freezing levels, in training from scratch mode and with and without data augmentation. The method was developed and tested using the AIDA public database. The best explored configuration showed in terms of performance: AUC of 98.6% and an accuracy of 93.9%.

The best configuration analyzed was compared with other state-of-the-art methods demonstrating an excellent ability to classify the fluorescence intensity in HEp-2 images. Unfortunately, in [34] a private database was used, so a more direct comparison on the same data is not possible. The difference in accuracy between our method and that of Zhou et al., even after optimization in accuracy, is still significant. Perhaps the best performance obtained in [34] is not due to a lower capacity of the CNN fine-tuning, but rather (in addition to the different database used) to the fact that, in [34], a segmentation of the images was performed in search of the cells, and starting from these, a clustering was performed that led to image classification. In this way, the method, effectively eliminating the background, shows greater robustness and, therefore, better performance.

The effectiveness of the fine-tuning technique was verified by comparing the performance of the same CNNs used by feature extractors (and coupled to SVM-type classifiers). With the same CNN used, fine-tuning has always given better results than the other method. It was possible to verify, for all the pre-trained networks analyzed, an, albeit, slight increase in performance with the use of the data augmentation method. Furthermore, as expected, for each network, the best performances were obtained with low freezing of the layers. The best performing network for the classification of fluorescence intensity in IIF images was SqueezeNet.

It should be noted that the CNNs tested in the various configurations obtained performances all contained in a very small range (about 3% of AUC).This denotes very similar classifying abilities (at least of the specific problem) of the CNN analyzed.

**Author Contributions:** V.T. developed the software, optimized the parameters, and helped to write the draft. D.C. conceived of the study, performed the statistical analysis, and drafted the manuscript. L.A. assisted in the statistical analysis and writing of the manuscript. G.R. participated in the design and coordination of the manuscript. All authors have read and agreed to the published version of the manuscript.

**Funding:** This research received no external funding.

**Conflicts of Interest:** The authors declare no conflict of interest.

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
