# Peer review of "Performance of Fine-Tuning Convolutional Neural Networks for HEp-2 Image Classification"

_applsci, doi:10.3390/app10196940_

Round 1

Reviewer 1 Report

Abstract – OK

Keywords

  • Spell out CNN, IIF, Hep-2? See discussion at: https://academia.stackexchange.com/questions/68985/using-acronyms-as-paper-keywords
  • Include ANA, AUC, ROC?

Introduction

  • Stronger first sentence needed as per comments in paper
  • Figure 1(a) – contains blanks? Is higher res needed?
  • Section 1.1 – should be a separate section: Literature Review
  • Tense in this section needs to be checked

2 Materials and methods

Lengthy descriptions

3 Results

Section 3.1 – should be part of section 3 only

Section 3 is short

4 Conclusions

No discussion section

GENERAL COMMENTS

Consistency in usage needed

  • Pre-trained or pretrained
  • K-fold or k-fold
  • Fine tuning or fine-tuning
  • Avoid use of European numbering 5.6 not 5,6
  • Features extractor or feature extractor

Errata – over 130

Related work by same authors

  • Cascio, D.; et al. Deep Convolutional Neural Network for HEp-2 Fluorescence Intensity Classification. Applied Sciences 2019, 9(3), 408.
  • Taormina, V.; Cascio, D.; Abbene, L.; Raso, G. HEp-2 intensity classification based on deep fine-tuning. In 7th International Conference on Bioimaging, BIOIMAGING 2020, Part of 13th International Joint Conference on Biomedical Engineering Systems and Technologies, BIOSTEC 2020 (pp. 143-149). SciTePress.

Summary

Paper is acceptable for publication after grammatical changes made. Literature survey appears to be up to date. Method and analysis appear to be sound, There is similarity to other recent work by the same authors although here they have used 4 CNNs not just one as in [29] although more are used in [13]

The results section is fairly brief and there could also be a discussion section.

Grammatical errors need to be corrected before publication

amended PDF file is attached for authors to check

Reviewer 2 Report

The paper addresses the interesting topic of Hep-2 images classification using CNN, with specific focus on signal processing aspects. Overall, the paper is clear and well written. However, I think that some improvements are still needed:

1) The introduction need to be expanded with (even coincise) references to other existing techniques in the Signal Processing literature that have been adopted for the task of image classification in diverse application contexts (including medical imaging). In particular, the Introduction delves too quickly on techniques based on images captured by cameras but misses some references about images built from (possibly pre-processed) radar signals, which represent another key methodology in the Signal Processing literature that also considers the adoption of CNN for classification. Prominent (quite recent) examples can be found, for instance, in the following paper about drone detection:

  • B. Taha et al: "A. Machine Learning-Based Drone Detection and Classification: State-of-the-Art in Research". IEEE Access 2019
  • A. Coluccia et al: Detection and Classification of Multirotor Drones in Radar Sensor Networks: A Review", Sensors, July 2020

but authors are strongly encouraged to consider also the references contained therein.

2) The section "Related work" can also be improved by adding a table that summarizes the main pros and cons of the referenced techniques. This would help a potential reader to easily follow the remaining of the manuscript.

3) A quantitative comparison in terms of computational complexity among the compared techniques is missing. As CNN-based classification (including the training phase) is known to be computationally intensive, highlighting the differences in terms of burden required by the reviewed methods could be useful to identify the most efficient techniques among them, shading at the same time some lights on the existing trade-off between the accuracy achieved an the cost in terms of computations. 

Round 2

Reviewer 2 Report

Authors correctly addressed all my comments.

This manuscript is a resubmission of an earlier submission. The following is a list of the peer review reports and author responses from that submission.

Round 1

Reviewer 1 Report

This paper proposes a fine-tuning method for the Convolutional Neural Network (CNN) used for the classification of fluorescence intensity in the positive/negative classes of Human Epithelial type 2 (HEp-2) images. The analysis was conducted using the four CNNs in the public database AIDA.

The details of the method are not described very clearly. Therefore, on the whole, it may be valuable for reference for researchers in the field, but for readers who are not familiar with CNN, it should be difficult to understand the method and its application.

I think it is helpful for the reader to understand the proposed method, if the author can take a CNN architecture as an example to illustrate which parameters in the network will affect the accuracy of discrimination, and how to adjust these parameters in this method for increasing the accuracy of discrimination.

Reviewer 2 Report

In the manuscript under review, a performance evaluation of four deep learning models has been presented. My comments on the work are as follows:

  • The novelty of the work is rather limited. The only contribution of this work is the comparison of the four existing deep learning architectures for automatic fluorescence classification in HEp-2 images, and the best performing architecture is claimed as proposed method. The reviewer would suggest to improve contribution of the paper and also clearly present the aim of the study. Many similar studies have been presented in the recent past that also discuss the deep learning for the said problem, few are mentioned in the manuscript related work too, however the additional contribution of this research, if any, is not clearly presented.
  • A similar study with fewer deep learning models was presented by the authors in BIOIMAGING 2020: Taormina, V., Cascio, D., Abbene, L. and Raso, G., 2020. HEp-2 intensity classification based on deep fine-tuning. In 7th International Conference on Bioimaging, BIOIMAGING 2020-Part of 13th International Joint Conference on Biomedical Engineering Systems and Technologies, BIOSTEC 2020 (pp. 143-149). SciTePress.
    This research may also be cited in the manuscript, and its difference with the present research must be highlighted.
  • There are some similarities of this research with: Cascio, D.; Taormina, V.; Raso, G. Deep Convolutional Neural Network for HEp-2 Fluorescence Intensity Classification. Appl. Sci. 2019, 9, 408.
    particularly the related work section and performance comparisons of deep learning architecture.
  • The related work is poorly written and needs significant improvements. Each piece of literature is described in one line paragraph. The continuity is missing, no pros and cons discussion of the related methods is presented. The motivation of the research is lacking.
  • In compared methods, the detection accuracy of Li method [20] is better than the proposed algorithm. Rejecting Li's method [20] by stating that it is using a smaller database is not convincing at all. The reviewer would suggest to evaluate the proposed method on the dataset used in Li [20] and present a fair comparison. Evaluating the performance on two large datasets would also increase the contributions of this research.
  • There are many grammatical and typographical errors in the paper. Few are mentioned below;
    • on page 8: "Table 3 show the best results obtained,..." must be "Table 3 shows the best results obtained,..."
    • Below Table 3, each sentence is written as a separate paragraph. This section must be revised and the results must be properly discussed.
    • on page 2: "In the IIF image analysis flow the first phase concerns the..." may be replaced with "In the IIF image analysis flow, the first phase concerns the..."

The research presented in this manuscript is not very novel, the paper is poorly written and is not very well-motivated.

Reviewer 3 Report

This work is focused on discriminating capabilities of pre-trained CNNs for intensity fluorescence classification in HEp-2 images. For this purpose, four well-known CNNs were analyzed (AlexNet, SqueezeNet, ResNet18 and GoogLeNet) by performing fine tuning with different modalities.

In a previous work from the same authors [19. Cascio, D., et al, 2019. Deep Convolutional Neural Network for HEp-2 Fluorescence Intensity Classification. Applied Sciences], they address the same problem of fluorescence intensity analysis using pre-trained CNNs (AlexNet, Sequeezenet, GoogleNET, VGG16-19, ResNet18-50-101 and Densenet201), which were used as feature extractors with a SVM classifier.

This manuscript uses the same database (AIDA) as in [19], the same pre-trained CNNs and a comparison with the same state-of-the-art methods: Di Cataldo [18]; Benammar [4]; and Cascio [19] (this new work also includes Li [20]). Besides, in both works the CNNs were evaluated as feature extractors in the pre-trained configuration and coupled to linear SVM classifiers.

In my opinion, the novelty and contribution of this paper seems not to be clear or be sufficient to be considered as an original contribution.